# Structure of substrate-bound SMG1-8-9 kinase complex reveals molecular basis for phosphorylation specificity

Lukas M Langer, Yair Gat, Fabien Bonneau, Elena Conti*

Department of Structural Cell Biology, Max Planck Institute of Biochemistry, Martinsried, Germany

**Abstract** PI3K-related kinases (PIKKs) are large Serine/Threonine (Ser/Thr)-protein kinases central to the regulation of many fundamental cellular processes. PIKK family member SMG1 orchestrates progression of an RNA quality control pathway, termed nonsense-mediated mRNA decay (NMD), by phosphorylating the NMD factor UPF1. Phosphorylation of UPF1 occurs in its unstructured N- and C-terminal regions at Serine/Threonine-Glutamine (SQ) motifs. How SMG1 and other PIKKs specifically recognize SQ motifs has remained unclear. Here, we present a cryo-electron microscopy (cryo-EM) reconstruction of a human SMG1-8-9 kinase complex bound to a UPF1 phosphorylation site at an overall resolution of 2.9 Å. This structure provides the first snapshot of a human PIKK with a substrate-bound active site. Together with biochemical assays, it rationalizes how SMG1 and perhaps other PIKKs specifically phosphorylate Ser/Thr-containing motifs with a glutamine residue at position +1 and a hydrophobic residue at position -1, thus elucidating the molecular basis for phosphorylation site recognition.

*For correspondence:
conti@biochem.mpg.de

Competing interests: The authors declare that no competing interests exist.

## Introduction

Family members of phosphatidylinositol 3-kinase-related kinases (PIKKs) activate distinct signaling pathways that promote cellular survival in different environmental and endogenous stress conditions (*Baretić and Williams, 2014*; *Imseng et al., 2018*; *Lempiäinen and Halazonetis, 2009*). Specifically, PIKKs oversee translation machinery in the cytoplasm (mTOR, SMG1), or regulate DNA damage repair in the nucleus (ATM, ATR and DNA-PK) (*Shimobayashi and Hall, 2014*; *Saxton and Sabatini, 2017*; *Yamashita, 2013*; *Yamashita et al., 2001*; *Blackford and Jackson, 2017*; *Elías-Villalobos et al., 2019*). With the exception of the enzymatically inactive TRAPP/Tra1, which serves as a scaffold in chromatin modification complexes, all other members of the PIKK family are Ser/Thr-protein kinases, and are among the largest proteins in the eukaryotic kinome. Recent publications have revealed the organization of PIKK active sites at better than 4 Å resolution (*Gat et al., 2019*; *Zhu et al., 2019*; *Yang et al., 2013*; *Jansma et al., 2020*; *Yates et al., 2020*), but the key question of how members of this kinase family recognize their substrates remains unanswered.

Human SMG1 is one of the largest PIKK family members (~410 kDa) and plays a crucial role in nonsense-mediated mRNA decay (NMD), a conserved pathway that regulates mRNA stability in the cytoplasm of eukaryotic cells (*Kurosaki and Maquat, 2016*; *Karousis and Mühlemann, 2019*). In its canonical surveillance function, the NMD pathway recognizes and degrades aberrant mRNAs containing premature translation termination codons, thus preventing the accumulation of truncated protein products. In addition, NMD also regulates the levels of a subset of normal, physiological transcripts, amounting to 5–10% of the transcriptome. In metazoans, SMG1 forms a stable complex with two additional proteins, SMG8 and SMG9, and specifically phosphorylates the RNA helicase UPF1. Phosphorylation of UPF1 is a crucial event in this pathway as it enables the recruitment of downstream NMD factors SMG5, SMG6 and SMG7, leading to ribonucleolytic cleavage of the RNA.

**eLife digest** The instructions for producing proteins in the cell are copied from DNA to molecules known as messenger RNA. If there is an error in the messenger RNA, this causes incorrect proteins to be produced that could potentially kill the cell. Cells have a special detection system that spots and removes any messenger RNA molecules that contain errors, which would result in the protein produced being too short.

For this error-detecting system to work, a protein called UPF1 must be modified by an enzyme called SMG1. This enzyme only binds to and modifies the UPF1 protein at sites that contain a specific pattern of amino acids – the building blocks that proteins are made from. However, it remained unclear how SMG1 recognizes this pattern and interacts with UPF1.

Now, Langer et al. have used a technique known as cryo-electron microscopy to image human SMG1 bound to a segment of UPF1. These images were then used to generate the three-dimensional structure of how the two proteins interact. This high-resolution structure showed that protein building blocks called leucine, serine and glutamine are the recognized pattern of amino acids. To further understand the role of the amino acids, Langer et al. replaced them one-by-one with different amino acids to see how each affected the interaction between the two proteins. This revealed that SMG1 preferred leucine at the beginning of the recognized pattern and glutamine at the end when binding to UPF1.

SMG1 is member of an important group of enzymes that are involved in various error detecting systems. This is the first time that a protein from this family has been imaged together with its target and these findings may also be relevant to other enzymes in this family. Furthermore, the approach used to determine the structure of SMG1 and the structural information itself could also be used in drug design to improve the accuracy with which drugs identify their targets.

SMG1 phosphorylates UPF1 specifically at Ser/Thr - Gln (SQ) motifs present in the unstructured N- and C-terminal regions that flank the helicase core (*Yamashita et al., 2001*; *Denning et al., 2001*). Specificity for a glutamine residue at the +1 position is shared by other PIKK family members, namely, ATM, ATR and DNA-PK kinases (*Kim et al., 1999*; *Bannister et al., 1993*). However, there is an additional layer of phosphorylation site specification. Among the 20 possible SQ motifs in UPF1, studies in vitro and in vivo have shown that only a selected few are effectively phosphorylated, including Ser1073, Ser1078, Ser1096 and Ser1116 (*Yamashita et al., 2001*; *Ohnishi et al., 2003*; *Durand et al., 2016*). Interestingly, these UPF1 phosphorylation sites share a Leu-Ser-Gln (LSQ) consensus sequence identical to the LSQ consensus motif identified in substrates of the ATM kinase (*O'Neill et al., 2000*; *Kim et al., 1999*). In this work, we studied the interaction between recombinant human SMG1-SMG8-SMG9 with UPF1 peptides using cryo-EM and mass spectrometry to identify the molecular basis with which SMG1, and potentially other PIKKs, recognizes specific phosphorylation sites in its substrate.

## Results and discussion

### Cryo-EM structure of the human SMG1-8-9 kinase complex bound to a UPF1 peptide

We used stably transfected HEK293T cells to express and purify a human wild-type SMG1-8-9 complex, as previously reported (*Gat et al., 2019*). The complex phosphorylated full-length recombinant UPF1 in a radioactive kinase assay (*Figure 1—figure supplement 1A*), confirming the enzymatic activity of purified SMG1 towards its physiological substrate. We selected a frequently phosphorylated site within UPF1 (*Yamashita et al., 2001*), Ser1078, and used a peptide spanning residues 1074–1084 (hereby defined as UPF1-LSQ) for subsequent structural and biochemical analysis (*Figure 1—figure supplement 1B*). We confirmed the ability of SMG1-8-9 to specifically phosphorylate UPF1-LSQ using a mass spectrometry-based phosphorylation assay (*Figure 1—figure supplement 1C and D*). This assay allowed us to monitor the relative amount of phosphorylation of a specific peptide over time. As a control, phosphorylation was abolished when Ser1078 was changed to Asp

(*Figure 1—figure supplement 1C and D*). Hence, the reconstitutions used in this study recapitulate specific phosphorylation site selection.

For structure determination, we incubated SMG1-8-9 with UPF1-LSQ and AMPPNP, a non-hydrolyzable ATP analogue, and subjected the sample to cryo-EM single particle analysis. The final reconstruction reached an overall resolution of 2.9 Å (*Figure 1—figure supplements 2* and *3*), and allowed us to further complete and refine the published model for SMG1-8-9 (*Supplementary file 1*; *Gat et al., 2019*). Briefly, SMG1 consists of an N-terminal solenoid 'arch' and a compact C-terminal 'head' region (*Figure 1A and B*). The C-terminal 'head' is formed by the tight interaction between the catalytic module, typical of Ser/Thr-kinases, and the so-called FAT and FATC domains (*Imseng et al., 2018*; *Baretić and Williams, 2014*; *Bosotti et al., 2000*). The N-terminal 'arch' provides binding sites for both SMG8 and SMG9 (*Figure 1A and B*). As we had previously reported, SMG9 contains an unusual G-fold domain that binds ATP rather than GTP or GDP (*Gat et al., 2019*).

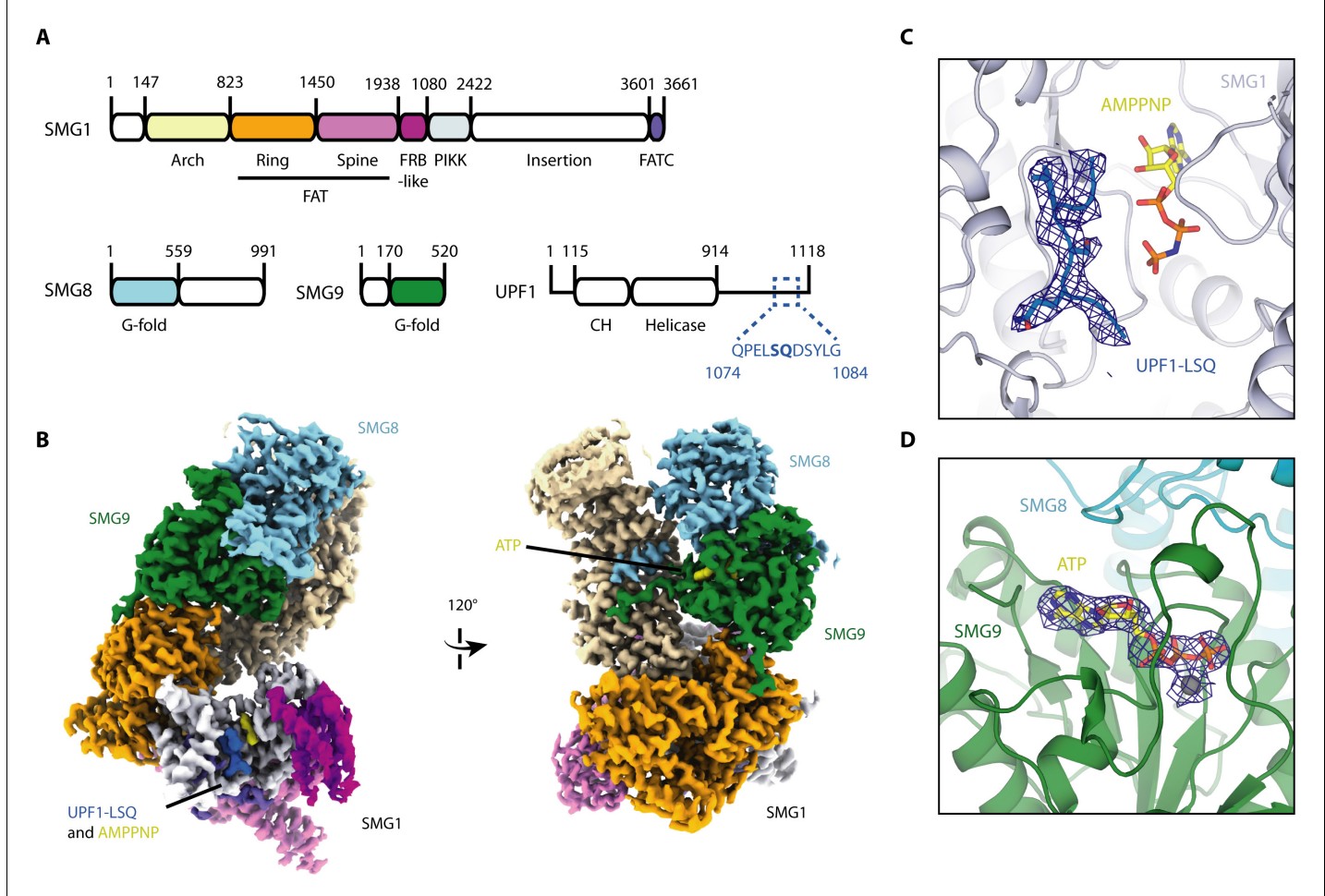

**Figure 1.** Cryo-EM reconstruction of SMG1-8-9 bound to UPF1-LSQ. (A) Domain organization of SMG1, SMG8, SMG9 and UPF1. White parts are not resolved in the reconstruction. The sequence and location of UPF1-LSQ is indicated with blue text and dotted lines. (B) Segmented cryo-EM reconstruction of substrate-bound SMG1-8-9. Two different views are shown; proteins and domains are colored as in A. (C) A zoomed-in view of SMG1 showing the kinase active site with bound AMPPNP and UPF1-LSQ. Reconstructed density for UPF1-LSQ is shown as a blue mesh. (D) Zoom-in showing ATP bound to SMG9 with reconstructed density displayed as a blue mesh.

The online version of this article includes the following figure supplement(s) for figure 1:

**Figure supplement 1.** SMG1-8-9 activity and UPF1 SQ motifs.
**Figure supplement 2.** Cryo-EM analysis of SMG1-8-9 bound to UPF1-LSQ.
**Figure supplement 3.** Cryo-EM data processing scheme.
**Figure supplement 4.** SMG9 is a G-fold containing protein binding ATP and exhibits distinct differences to the *bona fide* GTPase RAS.
**Figure supplement 5.** Quality of the reconstructed density.

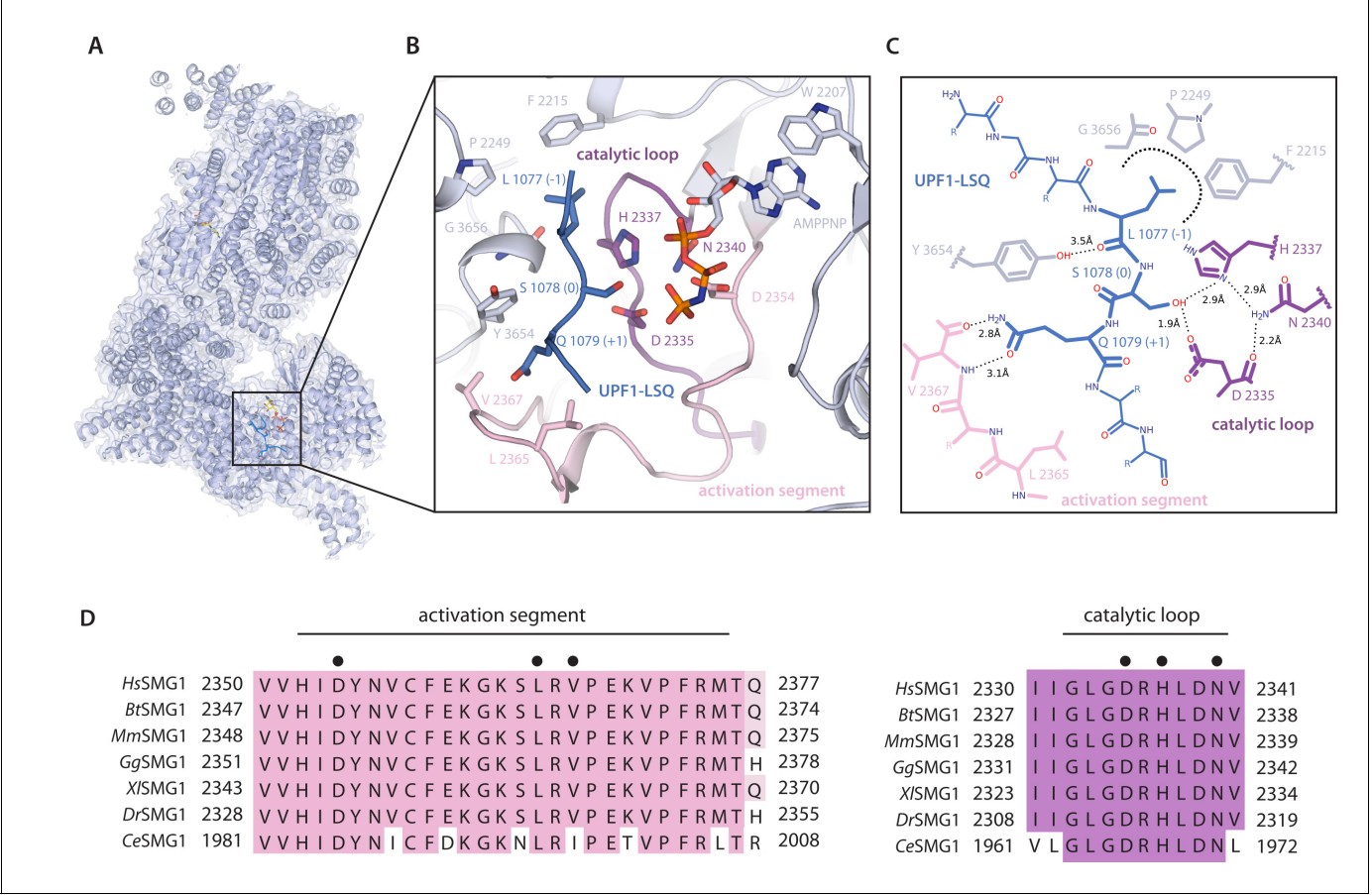

**Figure 2.** Organization of the substrate-bound kinase active site. (**A**) The structure of the entire complex is shown overlayed with transparent reconstructed density. The black box indicates the location of the kinase active site. (**B**) SMG1 active site with important residues shown as sticks. Activation segment and catalytic loop as indicated; the p-loop was omitted for clarity. UPF1-LSQ is shown in blue with positions of important residues highlighted. (**C**) Two-dimensional sketch of the SMG1 active site with key kinase-substrate interactions indicated. (**D**) Activation segment (pink) and catalytic loop (magenta) regions of a SMG1 sequence alignment are shown, indicating a high level of conservation across *Homo sapiens*, *Bos taurus*, *Mus musculus*, *Gallus gallus*, *Xenopus laevis*, *Danio rerio* and *Caenorhabditis elegans*. Key residues shown in B are highlighted by a black dot and activation segment and catalytic loop are indicated.

The online version of this article includes the following figure supplement(s) for figure 2:

**Figure supplement 1.** Comparison of SMG1 substrate-bound active site to other protein kinases.

**Figure supplement 2.** Recognition of UPF1-LSQ phospho-acceptor residue Ser1078 by SMG1 catalytic loop residues.

The local resolution of around 3 Å allowed us to model SMG9-bound ATP in the reconstructed density, revealing the molecular basis for how the adenosine nucleotide is recognized by this unusual G-fold domain (*Figure 1B and D*, *Figure 1—figure supplement 4*). Briefly, the G4 and G5 motifs responsible for the recognition of the base have rearranged to preferentially bind an adenine base rather than a guanine (*Figure 1—figure supplement 4*).

Importantly, compared to the previously published apo-SMG1-8-9 structure (*Gat et al., 2019*), the current reconstruction revealed extra density in the kinase active site accounting for both AMPPNP and UPF1-LSQ (*Figure 1B and C*, *Figure 1—figure supplement 5A,B and C*).

## Positioning of UPF1 Ser1078 in the SMG1 active site for phosphoryl transfer

The active site of SMG1 shows excellent density for residues 1075–1081 of UPF1-LSQ (*Figure 1C*, *Figure 1—figure supplement 5*). The directionality of the bound substrate peptide is consistent with that of other kinase structures, such as CDK2, and the arrangement of key active site residues is well conserved (*Figure 2—figure supplement 1*; *Bao et al., 2011*). The geometry of the catalytic

loop (residues 2332 to 2340) and of the activation segment (residues 2352 to 2375) in the SMG1 kinase domain as well as orientation of important active site residues are very similar to those observed in mTOR (*Figure 2—figure supplement 1B*), indicative of an active kinase state (*Yang et al., 2013*). Specific recognition of the phosphorylation site is achieved via conserved residues contributed by the activation segment and the catalytic loop as well as by the FATC domain (*Figure 2*). The hydroxyl group of Ser1078, the phospho-acceptor residue in UPF1-LSQ, is positioned by residues of the catalytic loop, in particular Asp2335 and His2337 (*Figure 2B and C*, *Figure 2— figure supplement 2A and B*). Consistent with the structural observations, mutation of either of the corresponding residues in SMG1, mTOR and other protein kinases results in catalytically inactive enzyme (*Bao et al., 2011*; *Madhusudan et al., 2002*; *Yang et al., 2013*; *Brown et al., 1995*; *Yamashita et al., 2001*; *Denning et al., 2001*). Therefore, the overall architecture of the substrate-bound SMG1 catalytic module corroborates the structural conservation among PIKK active sites and reveals that positioning of the substrate phospho-acceptor is achieved by residues that are shared between a wide range of protein kinases.

## Crucial recognition of a glutamine residue at +1 position of the UPF1 consensus motif

A glutamine residue following the phospho-acceptor site is the minimal requirement for UPF1 phosphorylation by SMG1 (*Figure 3A*; *Yamashita et al., 2001*). To validate the importance of this residue, we performed a mass spectrometry-based phosphorylation assay using a series of peptides based on UPF1-LSQ. We changed the residue at position +1 in the UPF1-LSQ peptide to test the effect of different side chain properties on phosphorylation. Only wildtype UPF1-LSQ was efficiently phosphorylated by SMG1 (*Figure 3B* and *Figure 1—figure supplement 1C*). In our structure, the glutamine at position +1 of UPF1-LSQ reaches into a hydrophobic cage formed by the SMG1 activation segment and FATC domain (*Figure 3C*). In particular, UPF1 Gln1079 stacks against Tyr3654 and Leu2365 and forms hydrogen bonds with the backbone of Val2367 (*Figure 3C*). The hydrophobic cage is highly conserved in other PIKKs that recognize SQ motifs (ATM, ATR and DNA-PK) but is different in mTOR (where Glu2369 is found at the equivalent position of SMG1 Leu2365) (*Figure 3D*). This difference is also apparent from the superposition of SMG1 with the 2.8 Å resolution structure of the ATM orthologue *Chaetomium thermophilum* (*Ct*) Tel1[ATM] and with mTOR (*Figure 2—figure supplement 1C and D*; *Jansma et al., 2020*; *Yang et al., 2013*). While the geometry of the hydrophobic cage is highly similar between SMG1 and *Ct*Tel1[ATM], it deviates in mTOR due to the described Leu to Glu substitution. Indeed, mTOR has been found to prefer small or non-polar residues at position +1 of its phosphorylation consensus motif (*Hsu et al., 2011*). Taken together, these observations provide a rationale for the difference in phosphorylation site specificity between SMG1, ATM, ATR, DNA-PK and mTOR. Intriguingly, the structural superposition with *Ct*Tel1[ATM] shows that its PIKK regulatory domain (PRD) places a Gln residue in the corresponding hydrophobic cage, effectively occupying the substrate Gln binding site (*Figure 2—figure supplement 1C*). This explains the autoinhibitory function of the ATM PRD domain (*Jansma et al., 2020*; *Yates et al., 2020*). The corresponding PRD domain in SMG1 is a ~ 1100 amino-acid long insertion (*Figure 1A*) that negatively impacts its kinase activity (*Deniaud et al., 2015*). However, there is no ordered density for this region in neither the previous apo-structure (*Gat et al., 2019*) nor in the current substrate-bound structure (*Figure 1A and B*).

## Preferred recognition of a leucine residue at −1 position of the UPF1 consensus motif

Previous results have indicated that SQ motifs preceded by a hydrophobic residue in position −1 are preferentially phosphorylated by SMG1 (*Yamashita et al., 2001*). In our model, the Leu residue at position −1 in the substrate forms a C-H···π-interaction with SMG1 Phe2215 and is further stabilized by hydrophobic interactions with SMG1 Pro2249 and Gly3656. The binding pocket is also restricted by the catalytic loop residues His2337 and Asp2339 (*Figure 4A*). To biochemically characterize the importance of position −1, we assayed a peptide library based on UPF1-LSQ, in which we varied the residue in position −1 to represent all those found in the 20 different SQ motifs of human UPF1. Following phosphorylation of the peptides over time, we could observe that SQ motifs carrying a hydrophobic residue in position −1 were more efficiently phosphorylated. Notably, a Leu

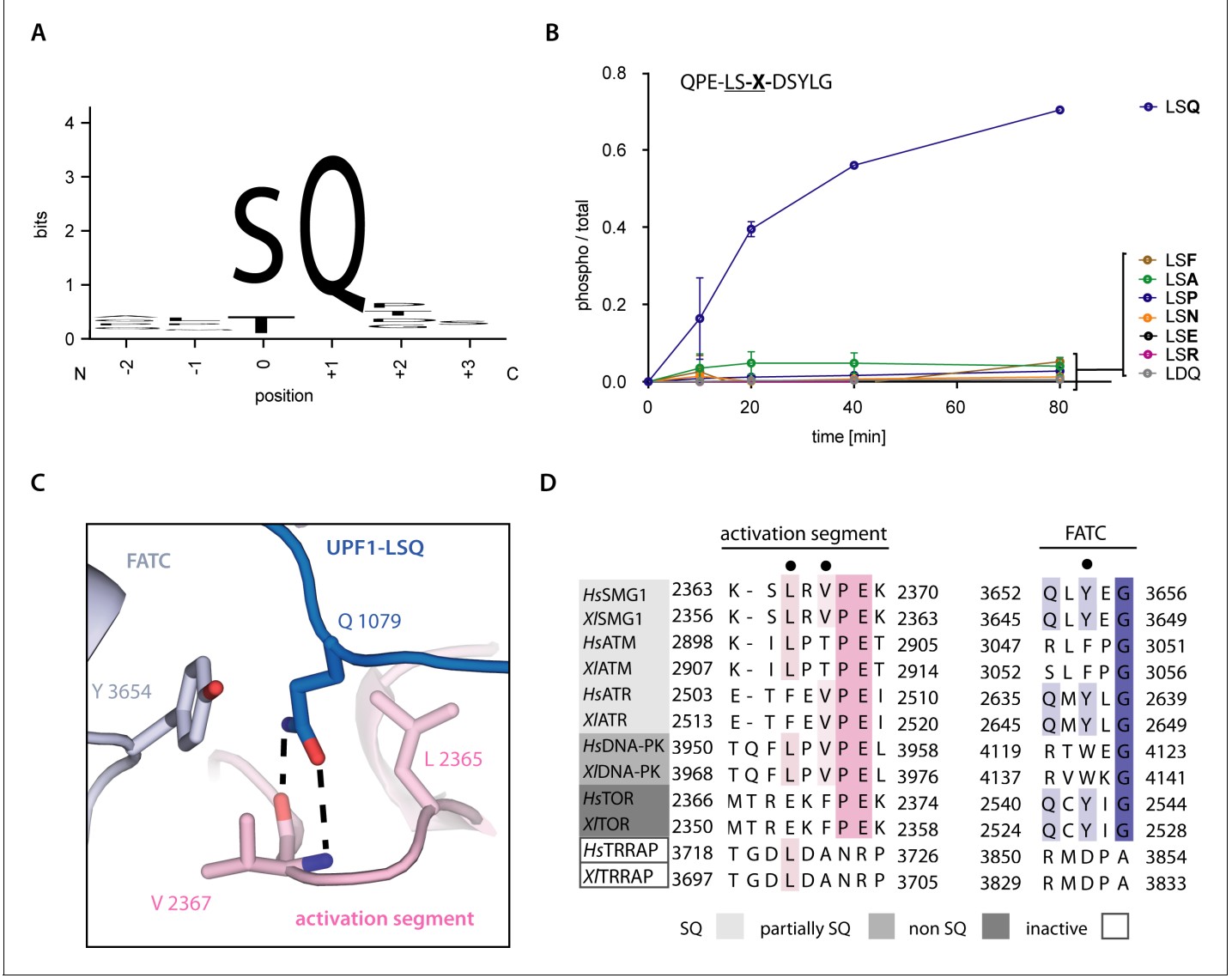

**Figure 3.** Recognition of position +1 glutamine of UPF1-LSQ. (**A**) Sequence logo derived from an alignment of all SQ motifs present in human UPF1 with the respective residue positions indicated. The heights of single letters correspond to the observed frequency at that position and the overall height of a stack of letters indicates the level of conservation (*Figure 1—figure supplement 1B* and *Figure 4—figure supplement 2*; *Crooks et al., 2004*). (**B**) Mass spectrometry-based phosphorylation assay comprising UPF1-LSQ and the indicated position +1 variations. The peptide sequence is indicated in the upper left with the varied position marked as 'X'. Error bars representing standard deviations calculated from independent experimental triplicates are shown. (**C**) Zoom-in of the SMG1 active site showing the recognition of UPF1 position +1 glutamine by SMG1 residues located in the activation segment and FATC domain. Residues of interest are shown as sticks. Colors as in *Figure 2*. (**D**) Alignment of PIKK sequences from *Homo sapiens* and *Xenopus laevis* with the activation segment and FATC domain sequences shown and colored according to conservation. PIKKs are grouped by phosphorylation site specificity and residues highlighted in subfigure C are indicated by a black dot.

elicited the highest phosphorylation rate (*Figure 4B*). An end-point measurement experiment using single peptides confirmed these observations (*Figure 4—figure supplement 1*). We conclude that a Leu at position −1 is optimal for the interaction with SMG1 at the structural level. This is reflected at the biochemical level, whereby decreasing hydrophobicity of the residue at the −1 position negatively affects phosphorylation efficiency.

Interestingly, further analysis of the final time points in the time course phosphorylation experiment showed that the SQ motifs that carry rather hydrophobic residues at the −1 position (and are therefore more efficiently phosphorylated) reside exclusively in the UPF1 C-terminus (*Figure 4C*, *Figure 4—figure supplement 2*). To validate our hypothesis on the importance of position −1

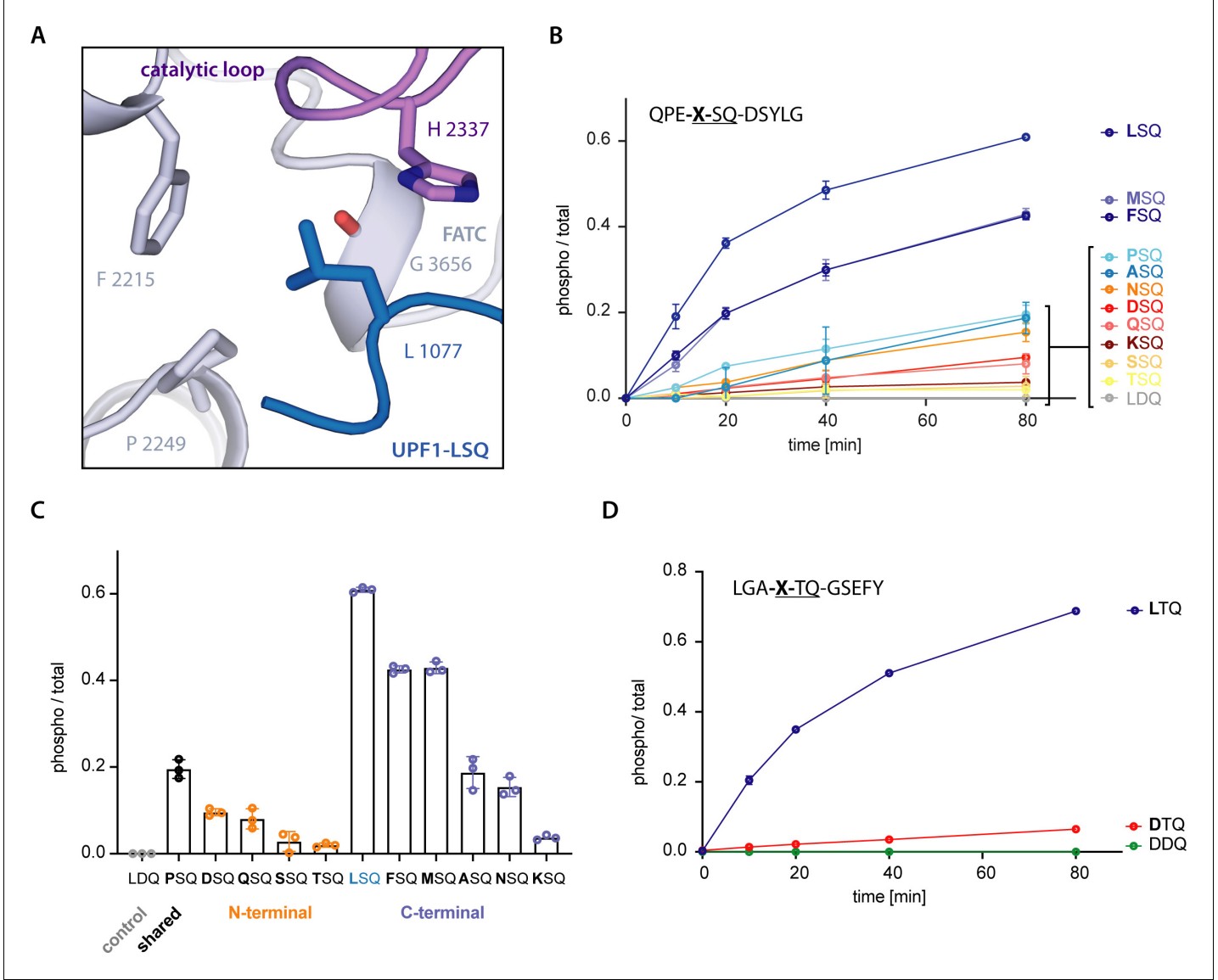

**Figure 4.** SMG1 preferentially selects for substrates with hydrophobic residues in position -1. (**A**) Recognition of position -1 residue of UPF1-LSQ by SMG1. Important residues are shown as sticks and colored as in *Figure 2*. (**B**) Mass spectrometry-based phosphorylation assay with UPF1-LSQ and derivatives varied in position -1. The peptide sequence is indicated in the upper left with the varying position marked as "X". Error bars representing standard deviations calculated from independent experimental triplicates are shown and curves are colored according to hydrophobicity of the position -1 residue (*Eisenberg et al., 1984*). The most hydrophobic peptides are in blue while non-hydrophobics are in red. (**C**) Final time points of experiment shown in B. Peptides grouped and colored according to the location of the respective position -1 residue in UPF1 N- or C-terminus. Individual data points are shown as circles and error bars representing standard deviations are indicated. (**D**) Mass spectrometry-based phosphorylation assay with UPF1 N-terminus phosphorylation site 28 and the indicated position -1 variation. The peptide sequence is shown in the upper left with the varied position marked as "X". Error bars represent standard deviations resulting from independent experimental triplicates. A tyrosine residue was added to the C-terminal end of the wildtype sequence to increase absorbance at $\lambda$ = 280 nm.

The online version of this article includes the following figure supplement(s) for figure 4:

**Figure supplement 1.** In vitro phosphorylation of UPF1-LSQ and derivatives.

**Figure supplement 2.** Alignment of UPF1 N- and C-terminal sequences from *Homo sapiens*, *Bos taurus*, *Canis lupus*, *Mus musculus*, *Gallus gallus*, *Xenopus tropicalis*, *Danio rerio* and *Caenorhabditis elegans*.

hydrophobicity, we turned to a known phosphorylation site in the N-terminus, Thr28. We tested whether SMG1-8-9 phosphorylation activity toward this motif could be enhanced by changing the residue at position −1 from the naturally occurring Asp to a Leu. Indeed, mutating the residue upstream of this SQ motif (Asp27Leu) resulted in a gain-of-function effect in the phosphorylation assay (*Figure 4D*). These findings are in agreement with data for ATM and ATR (*Kim et al., 1999*), although the residues involved in the recognition of UPF1 Leu1077 have diverged, suggesting that the details of −1 recognition will differ in other PIKKs. Finally, we do not observe extensive interactions between SMG1 and the peptide residues preceding or following the LSQ motif in our structure. Consistently, we did not detect a marked effect on phosphorylation in a time course experiment where we changed the residue at position +2 of UPF1-LSQ (*Figure 4—figure supplement 1B*).

## Conclusions

In this manuscript, we report the first structure of a substrate-bound PIKK active site, thereby revealing the basis for phosphorylation site selection by SMG1 and other PIKK family members. The results elucidate the mechanism of phospho-acceptor recognition, and explain the specificity for Ser-containing substrates with a glutamine downstream residue at position +1 and an upstream hydrophobic residue at position −1 (particularly Leu). These findings can be extrapolated to other PIKK members, such as ATM and ATR, and suggest a specific mechanism for PRD function by acting as a pseudosubstrate. Our results provide molecular insights into a key step of the NMD pathway. Whether phosphorylation of full-length UPF1 by SMG1 involves additional elements of recognition and/or additional levels of regulation will be a subject for future studies.

# Materials and methods

**Key resources table**

| Reagent type (species) or resource | Designation | Source or reference | Identifiers | Additional information |
|---|---|---|---|---|
| Gene (*Homo sapiens*) | SMG1 | Shigeo Ohno lab | Uniprot Q96Q15 | |
| Gene (*Homo sapiens*) | SMG8 | Shigeo Ohno lab | Uniprot Q8ND04 | |
| Gene (*Homo sapiens*) | SMG9 | Shigeo Ohno lab | Uniprot Q9H0W8 | |
| Cell line (*Homo sapiens*) | HEK293T | ATCC | | |
| Strain, strain background (*Escherichia coli*) | BL21 Star (DE3) pRARE | EMBL Heidelberg Core Facility | | Electrocompetent cells |
| Peptide, recombinant protein | UPF1- LSQ (peptide 1078) and derivatives, UPF1-peptide 28 and derivatives | in-house as described in the Materials and methods section | | |
| Chemical compound, drug | AMPPNP | Sigma-Aldrich | | |
| Chemical compound, drug | ATP | Sigma-Aldrich | | |
| Software, algorithm | RELION | DOI: 10.7554/eLife.42166 | RELION 3.0 | |
| Software, algorithm | Cryosparc | DOI: 10.1038/nmeth.4169 | Cryosparc2 | |
| Software, algorithm | CtfFind | DOI: 10.1016/j.jsb.2015.08.008 | CtfFind4.1.9 | |
| Software, algorithm | Cryosparc | DOI: 10.1038/nmeth.4169 | Cryosparc2 | |

*Continued on next page*

*Continued*

| Reagent type (species) or resource | Designation | Source or reference | Identifiers | Additional information |
|---|---|---|---|---|
| Software, algorithm | UCSF Chimera | UCSF, https://www.cgl.ucsf.edu/chimera/ | | |
| Software, algorithm | UCSF ChimeraX | UCSF, https://www.rbvi.ucsf.edu/chimerax/ | | |
| Software, algorithm | COOT | http://www2.mrc-lmb.cam.ac.uk/personal/pemsley/coot/ | | |
| Software, algorithm | PHENIX | https://www.phenix-online.org/ | PHENIX 1.17 | |
| Software, algorithm | PyMOL | PyMOL Molecular Graphics System, Schrodinger LLC | PyMOL 2.3.2 | https://www.pymol.org/ |

## Protein expression and purification

The SMG1-SMG8-SMG9 complex was expressed and purified as previously described (*Gat et al., 2019*). Briefly, a pool of HEK293T cells (obtained from ATCC) stably expressing full length SMG1 (N-terminally fused to a TwinStrep-tag and a 3C protease cleavage site), SMG8 and SMG9 was established using the *piggybac* method by initially transfecting the cells using polyethylenimine (*Yusa et al., 2011*; *Li et al., 2013*). The source cells were authenticated by genotyping (Eurofins) and tested negative for mycoplasma contamination (LookOut Mycoplasma PCR Detection Kit, Sigma-Aldrich). For SMG1-SMG8-SMG9 expression, cultures were adjusted to a density of $1 \times 10^6$ cells per mL in FreeStyle 293 Expression Medium (Gibco, Thermo Fisher Scientific). The cells were induced by addition of doxycycline and were harvested 48 hr after induction. After lysis by douncing the cells in 1xPBS, 1 mM $MgCl_2$ and 1 mM DTT supplemented with DNase I, Benzonase and EDTA-free cOmplete Protease Inhibitor Cocktail (Roche) the cleared lysate was applied to a StrepTrap HP column (Sigma-Aldrich) and the complex affinity purified using the N-terminal TwinStrep-tag on SMG1. After washing with 50 column volumes of 1xPBS, 1 mM $MgCl_2$ and 1 mM DTT the complex was eluted using wash buffer supplemented with 2.5 mM desthiobiotin. SMG1-8-9 was further purified by size-exclusion chromatography using a Superose 6 Increase 10/300 GL column (Sigma-Aldrich) equilibrated with 1xPBS, 1 mM $MgCl_2$ and 1 mM DTT (Aekta purifier FPLC system, GE Healthcare). Purified SMG1-8-9 was concentrated up to 6 µM and stored in gel filtration buffer. To obtain full-length unphosphorylated human UPF1, the protein was expressed in *Escherichia coli* BL21 STAR (DE3) pRARE fused to a C-terminal 6xHis-tag cleavable with Tobacco etch virus (TEV) protease, as described before (*Chakrabarti et al., 2011*; *Chakrabarti et al., 2014*). Bacteria were grown at 37°C in TB medium shaking at 180 rpm and induced using IPTG at an OD of 2 for overnight expression at 18°C. Harvested bacteria (6000 rpm, 10 min) were lysed by sonication in lysis buffer (50 mM Tris-Cl pH 7.5, 500 mM NaCl, 10 mM Imidazole, 1 mM β-mercaptoethanol, 10% (v/v) glycerol, 2 mM $MgCl_2$ and 0.2% (v/v) NP-40) supplemented with DNase I and EDTA-free cOmplete Protease Inhibitor Cocktail (Roche). The lysate was cleared by centrifugation (25.000 rpm, 30 min), filtered and combined with TALON resin (Takara) equilibrated with lysis buffer for gravity-flow affinity purification. After washing with 70 column volumes of lysis buffer, the protein was eluted with lysis buffer supplemented with 300 mM imidazole pH 7.5 and the eluate was combined with His-tagged TEV protease and dialyzed overnight against 20 mM HEPES pH 7.5, 85 mM KCl, 1 mM $MgCl_2$, 10% (v/v) glycerol and 2 mM DTT. The dialyzed sample was passed over another TALON column by gravity-flow, in order to separate cleaved protein from the cleaved-off His-tag, the His-tagged TEV protease and uncleaved UPF1 protein. The flow-through of this column contained cleaved UPF1 and was loaded on a HiTrap Heparin HP column (GE Healthcare). Following binding and washing with Heparin buffer A (as for dialysis), UPF1 was eluted by a gradient increasing salt concentration from 85 mM to 500 mM over 50 column volumes (Aekta prime FPLC system, GE Healthcare). The peak corresponding to full-length UPF1 was pooled and concentrated before a final sizing step using a Superdex 200 Increase 10/300 GL column (Sigma-Aldrich) equilibrated with Heparin buffer A (Aekta purifier FPLC system, GE Healthcare). Purified full-length UPF1 was pooled and concentrated up to 30 µM using an Amicon Ultra Centrifugal Filter (50 kDa MWCO, Merck). All described protein

purification steps were carried out at 4°C and all purified proteins were flash frozen in size-exclusion buffer using liquid nitrogen and stored at −80°C until further usage.

## Cryo-EM sample preparation and data collection

A sample of 0.5 µM (final concentration) purified SMG1-SMG8-SMG9 was mixed with 0.5 mM of UPF1-LSQ, 1 mM AMPPNP, 2 mM MgCl$_2$, 2 mM DTT and 0.04% (v/v) n-octyl-beta-D-glucoside in 1xPBS and incubated for 30 min on ice. The UPF1-LSQ peptide (sequence: QPELSQDSYLG) was synthesized in-house as described for the mass spectrometry-based phosphorylation assay. A 4 µL sample was applied to a glow-discharged Quantifoil R1.2/1.3, Cu 200 mesh grid and incubated for 30 s at 4°C and approximately 100% humidity. Grids were subsequently plunge frozen directly after blotting using a liquid ethane/propane (37% ethane, temperature range when plunging: −170°C to −180°C) mixture and a ThermoFisher FEI Vitrobot IV set to a blot time of 3.5 s and a blot force of 4. Cryo-EM data were collected using a ThermoFisher FEI Titan Krios microscope operated at 300 kV equipped with a post-column GIF (energy width 20 eV) and a Gatan K3 camera operated in counting mode, the SerialEM software suite, and a beam-tilt based multi-shot acquisition scheme. Movies were recorded at a nominal magnification of 81.000x corresponding to a pixel size of 1.094 Å at the specimen level. The sample was imaged with a total exposure of 68.75 e$^-$/Å$^2$ evenly spread over 5.5 s and 79 frames. The target defocus during data collection ranged between −0.8 and −2.9 µm.

## Cryo-EM data processing

Data processing was carried out using RELION 3.0 (*Zivanov et al., 2018*) unless stated otherwise. Beam-induced sample motions were corrected and dose-weighting was carried out using the RELION implementation of MotionCor2 (*Zheng et al., 2017*). Particles were picked using Gautomatch (https://www.mrc-lmb.cam.ac.uk/kzhang/Gautomatch/) and CTF estimation was done using the RELION wrapper for CtfFind4.1 (*Rohou and Grigorieff, 2015*). After extraction (box size: 320 pix, 1.094 Å/pix) and downsampling (box size: 80 pix, 4.376 Å/pix), 4,368,586 particles were submitted to two rounds of reference-free 2D classification. A subset of the cleaned candidate particles was used to generate an initial model using CryoSPARC v2 (*Punjani et al., 2017*). 1,524,355 selected particles were 3D classified before re-extracting 886,714 particles with original sampling followed by two additional rounds of 3D classification resulting in 481,754 final particles. All classification steps were carried out with the total amount of particles being distributed over multiple batches. After 3D auto-refinement, sharpening (b-factor = −119.5) and Ctf refinement in RELION 3.0, the final refined map (3D auto-refinement) was again submitted to RELIONs' post-processing routine for automatic B-factor weighting and high-resolution noise substitution (b-factor = −102.6). The final reconstruction (EMD-11063) reached an overall resolution of 2.9 Å with local resolution ranging from 2.8 Å to 4.5 Å as estimated by RELION 3.0.

## Model building and refinement

The reconstructed density was interpreted using COOT (version 1.0) and our previously published model of SMG1-8-9 (PDB: 6SYT) (*Emsley et al., 2010*). Model building was iteratively interrupted by real-space refinements using PHENIX (version 1.17) (*Adams et al., 2010*; *Liebschner et al., 2019*). Statistics assessing the quality of the final model (PDB ID 6Z3R) were generated using Molprobity (*Chen et al., 2010*; *Supplementary file 1*). FSC curves were calculated using PHENIX and the 3D FSC online application (*Tan et al., 2017*). Images of the calculated density and the built model were prepared using UCSF Chimera (*Pettersen et al., 2004*), UCSF ChimeraX (*Goddard et al., 2018*) and PyMOL (version 2.3.2).

## Radioactive in vitro kinase assay

In vitro kinase assays were essentially carried out as before (*Gat et al., 2019*). 1 µM of full-length UPF1 was mixed with 50 nM SMG1-8-9, 10mM MgCl$_2$ and 2mM DTT in 1xPBS. The reaction was started by adding 0.5mM ATP and 0.06 µM of $\gamma$-$^{32}$P-labeled ATP. The reaction was incubated at 30°C and samples were taken at different time points to follow phosphorylation over time. The samples were immediately quenched by adding SDS-containing sample buffer and initially analyzed by SDS gel electrophoresis followed by Coomassie-staining. Phosphoproteins were subsequently detected using autoradiography and a Typhoon FLA7000 imager (GE Healthcare).

## Mass spectrometry-based in vitro kinase assay

All peptides were synthesized in-house using solid-phase peptide synthesis and the quality of the product was assessed by electrospray ionization mass spectrometry (ESI MS). For the purpose of this study, peptides were dissolved in 1xPBS supplemented with 500 mM HEPES pH 7.4. Two types of experiments were carried out. Firstly, several peptides (typically comprising a library) and a control were mixed, and their individual phosphorylation ratios were determined at several time points (0, 10, 20, 40 and 80 min). Secondly, one end-point measurement experiment was carried out. In this setup, a single peptide was mixed with a control and the relative phosphorylation ratio was determined at a single, final time point (80 min). This type of experiment was used to assess whether effects on phosphorylation ratios observed in time course assays are caused by competition between several peptides for SMG1 binding. In both cases, 0.5 µM of kinase complex was combined with 0.1 mM of each peptide, 0.5 mM ATP, 20 mM $MgCl_2$ and 2 mM DTT in 1xPBS. The reaction was started by addition of kinase and incubated at 30°C. Samples were taken at desired time points and immediately quenched after collection by adding EDTA to a final concentration of 50 mM on ice.

In order to remove kinase complex and transfer the peptides into a compatible buffer, we made use of home-made StageTips (*Rappsilber et al., 2007*). Poly(styrenedivinylbenzene)copolymer (SDB-XC) was washed with methanol by centrifugation before being washed again with buffer B (0.1% (v/v) formic acid, 80% (v/v) acetonitrile). Buffer A (0.1% (v/v) formic acid) was used for equilibration of the SDB-XC material. Following sorbent equilibration, the sample was applied and the tips were washed using buffer A. Finally, the sample was eluted using buffer B. Using an Agilent 1290 HPLC, typically about 5 µL of the sample in 70% (v/v) acetonitrile and 0.05% (v/v) trifluoroacetic acid were flow-injected (250 µL/min) into a Bruker maXis II ETD mass spectrometer for ESI MS time-of-flight analysis. Peptides were ionized at a capillary voltage of 4500 V and an end plate offset of 500 V. Full scan MS spectra (200–1600 m/z) were acquired at a spectra rate of 1 Hz and a collision energy of 10 eV. All experiments were carried out as independent experimental triplicates. Raw data files were processed using Bruker Compass DataAnalysis software. The m/z spectra were deconvoluted by maximum entropy with an instrument resolving power of 10,000. The $^{12}$C peaks corresponding to individual peptides were identified in the resulting neutral spectra and integrated, both for masses accounting for unphosphorylated and phosphorylated peptides. To calculate a relative phosphorylation ratio, the area for phosphorylated peptide was divided by the sum of phosphorylated and unphosphorylated peptide. All time points were normalized to time point 0. Means of independent experimental triplicates and error bars indicating standard deviations were visualized using Prism (GraphPad).

## Acknowledgements

We thank Daniel Bollschweiler and Tillman Schäfer at the MPIB cryo-EM facility for help with EM data collection and Elisabeth Weyher and Stefan Pettera at MPIB biochemistry core facility at MPIB for conducting mass spectrometry and synthesis of peptides used in this study, respectively. We thank Christian Benda and J Rajan Prabu for maintenance and development of computational infrastructure and Daniela Wartini for assistance in mammalian tissue culture. We are grateful to Courtney Long and members of the group for input and discussion on the manuscript. This study was supported by funding from the Max Planck Gesellschaft, the European Commission (ERC Advanced Investigator Grant EXORICO), and the German Research Foundation (DFG SFB1035, GRK1721, SFB/TRR 237) to EC and a Boehringer Ingelheim Fonds PhD fellowship to LL.

## Additional information

### Funding

| Funder | Grant reference number | Author |
| --- | --- | --- |
| Boehringer Ingelheim Fonds | PhD fellowship | Lukas M Langer |
| Max-Planck-Gesellschaft | | Elena Conti |
| European Commission | ERC Advanced Investigator Grant EXORICO | Elena Conti |

| Deutsche Forschungsge-meinschaft | SFB1035 | Elena Conti |
| Deutsche Forschungsge-meinschaft | GRK1721 | Elena Conti |
| Deutsche Forschungsge-meinschaft | SFB/TRR 237 | Elena Conti |

The funders had no role in study design, data collection and interpretation, or the decision to submit the work for publication.

### Author contributions
Lukas M Langer, Conceptualization, Data curation, Formal analysis, Validation, Investigation, Visualization, Methodology, Writing - original draft, Writing - review and editing; Yair Gat, Conceptualization, Validation, Investigation, Visualization, Writing - review and editing; Fabien Bonneau, Formal analysis, Validation, Methodology, Writing - review and editing; Elena Conti, Conceptualization, Resources, Supervision, Funding acquisition, Writing - original draft, Project administration, Writing - review and editing

### Author ORCIDs
Lukas M Langer [ID] https://orcid.org/0000-0002-9977-2427
Yair Gat [ID] https://orcid.org/0000-0002-2338-9384
Fabien Bonneau [ID] http://orcid.org/0000-0001-8787-7662
Elena Conti [ID] https://orcid.org/0000-0003-1254-5588

### Decision letter and Author response
Decision letter https://doi.org/10.7554/eLife.57127.sa1
Author response https://doi.org/10.7554/eLife.57127.sa2

## Additional files
### Supplementary files
- Supplementary file 1. Cryo-EM data collection, refinement and validation statistics.
- Transparent reporting form

### Data availability
EM data have been deposited in EMDB under the accession code EMD-11063. The model has been deposited in PDB under the accession 6Z3R.

The following datasets were generated:

| Author(s) | Year | Dataset title | Dataset URL | Database and Identifier |
| --- | --- | --- | --- | --- |
| Langer LM, Gat Y, Conti E | 2020 | Structure of SMG1-8-9 kinase complex bound to UPF1-LSQ | https://www.ebi.ac.uk/pdbe/entry/emdb/EMD-11063 | Electron Microscopy Data Bank, EMD-11063 |
| Langer LM, Gat Y, Conti E | 2020 | Structure of SMG1-8-9 kinase complex bound to UPF1-LSQ | https://www.rcsb.org/structure/6Z3R | RCSB Protein Data Bank, 6Z3R |

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
