## [Decision Letter]

**Acceptance summary:**

This study reports a high resolution structure using CryoEM of a PIKK family kinase in complex with a peptide substrate and an ATP analog. It is the first member of this family of protein kinases to be captured at atomic resolution with a peptide substrate. The structural interactions between enzyme and substrate were corroborated with very solid biochemical analysis including a range of kinase assays with various peptides. This work will serve as a model for related kinases including ATM, ATR, DNAPK, and mTOR.

**Decision letter after peer review:**

Thank you for submitting your article "Structure of substrate-bound SMG1-8-9 kinase complex reveals molecular basis for phosphorylation specificity" for consideration by *eLife*. Your article has been reviewed by Reviewing/Senior Editor Philip Cole and three reviewers. The following individuals involved in review of your submission have agreed to reveal their identity: Nikolaus Grigorieff (Reviewer #1); Kacper Rogala (Reviewer #2); Michael B Yaffe (Reviewer #3).

The reviewers have discussed the reviews with one another and the Reviewing Editor has drafted this decision to help you prepare a revised submission.

We would like to draw your attention to changes in our revision policy that we have made in response to COVID-19 (https://elifesciences.org/articles/57162). Specifically, we are asking editors to accept without delay manuscripts, like yours, that they judge can stand as *eLife* papers without additional data, even if they feel that they would make the manuscript stronger. Thus the revisions requested below, although fairly numerous, are meant to address clarity and presentation.

Summary:

Langer et al. elucidate a cryo-EM structure of a human PIKK family member SMG-1-8-9 in complex with a short peptide substrate (from UPF1) and a non-hydrolyzable analog of ATP, AMPPNP. This is an extension of the cryo-EM work that they did in 2019 on apo SMG-1-8-9 (Gat et al., 2019). However, this study is in fact more interesting than the previous work because it captures the SMG1 kinase in a state with its substrate peptide bound, frozen in time just before the actual act of phosphorylation. Importantly, the authors do an extensive characterization of the peptide recognition sequence, and reveal the chemical compatibility of the LSQ sequence and its derivatives for phosphorylation by SMG1. They compare the binding site of SMG1 to other members of the PIKK family and explain the observed differences between them in terms of substrate recognition. Overall, we believe that this work is novel and interesting. In general, the claims are supported by solid data, and the structure is validated with substantial biochemical work. Although we do not think additional experimental work is needed, below we make the following suggestions for revisions to strengthen the manuscript.

Essential revisions:

-The structure and its interpretation seem plausible. However, we wonder if using an 11-amino acid peptide as a substrate captures all the contacts relevant for the kinase specificity that the authors are interested in. Is it possible that the full UPF1 molecule makes tertiary contacts with the kinase that are important for specificity? The authors should consider this possibility and explain why they can rule this out. Related to the above point, the authors state that "there are no extensive interactions between SMG1 and the residues preceding or following the LSQ motif in our structure." It is not clear how they can conclude this from their structure. If this is known from other work, they should cite it.

-The authors seem to have side-stepped the other 3D classes that were generated during data processing. There are at least two classes (center top and center bottom in Figure 1—figure supplement 3) that show extra density that is not present in the final reconstruction presented. That density appears to extend from SMG8's stalk and reach towards the FRB of SMG1. According to Li et al., 2019), this extra density is likely the kinase inhibitory domain (KID) of SMG8. This density also appeared in the authors' previous map of the apo SMG-1-8-9 complex (EMDB 10348), but they avoided discussing it in their previous manuscript (Gat et al., 2019). If the authors feel that delving into this would be too speculative, adding a label and a note to point out what they represent would be helpful.

-A discussion regarding nucleotide binding to SMG9 and its effects on SMG8 binding and SMG1 catalytic activity is lacking. The authors in their previous work revealed a GTP/GDP nucleotide binding pocket in *C. elegans* SMG1 (Li et al., 2017), and then realized that human SMG9 co-purifies with bound ATP instead of GTP (Gat et al., 2019). Can human SMG1 also associate with GTP/GDP? In Figure 1—figure supplement 4 of this manuscript, the authors talk extensively about the adaptations for adenine vs guanine binding. This topic appears to be rather confusing in the field, so please include a short discussion that deals with the nucleotide-binding matter, including the mutagenesis study from another SMG-1-8-9 structure paper by Li et al., 2019.

-The methods are not sufficiently detailed, and it would be challenging to reproduce the authors' work by simply following them.

-Figure 3A and Figure 3 legend, particularly the statement that sequence logo letter size reflects the frequency of occurrence is not technically correct. True sequence logos use bit scores that typically do not reflect the frequency of occurrence but rather indicate the information content (in bits) that is being provided by each residue in that position in terms of the informational entropy content in each position of a motif, at least as originally described by Tom Schneider and Mike Stephens in 1990. A good review of the concept is found in Crooks et al., 2004. (http://www.genome.org/cgi/doi/10.1101/gr.849004). The authors should clarify how they are using the sequence logos here.

-Regarding the data in Figure 3B described in subsection “Crucial recognition of a glutamine residue at +1 position of the UPF1 consensus motif”, the claims of "peptide library" and "systematically" in the text go a bit beyond the data, since they are really looking at a small series of peptides, not a complete library of all residues in that position. The data is solid, and there is no need to do anything more, other than tone down the claims of this being a 'library'. We think the use of the term 'library' in Figure 4 and its description is fine, since in that case the authors really are looking at the set of all possible X-SQ sites that exist in UPF1.

-The authors may want to expound a bit more on the similarity and differences in the hydrophobic cage between SMG1 and other PIKKs, particularly the Leu to Glu activation loop substitution seen in mTor (Figure 2—figure supplement 1B). This is important because the optimal kinase motif for mTor is NOT SQ as it is in ATM, ATR and DNA-PK, but rather more like SF or SP (see Figure 2 in Hsu et al., 2011).

-Subsection “Preferred recognition of a leucine residue at -1 position of the UPF1 consensus motif” and Figure 4C- presumably the point the authors are making is that C-terminal SQ motifs are better phosphorylated than the N-terminal ones because they are better matches to the hydrophocis-Ser-Gln motif. Perhaps they can state this conclusion a bit more clearly.

-Similarly, it appears from Figure 2C that the Gln side chain is actually helping stabilize the activation segment conformation. Is this correct, and if so, shouldn't this be alluded to somewhere in the Results section?

-Regarding the catalytic mechanism, as the authors have a (not so common) ternary complex protein kinase crystal structure containing both a peptide substrate and a nucleotide substrate, it would be nice to say a bit more about the relevance to potential phosphoryl transfer transition states. There has been a classical discussion in the field among physical organic chemists and enzymologists about whether phosphoryl transfers involving monoesters like the γ phosphate of ATP are more associative or dissociative. High resolution structures have been used to inform these discussions. For examples, please see: PMID:9408938, PMID:21513457, PMID:25399640

---

## [Author Response]

Summary:Langer et al., elucidate a cryo-EM structure of a human PIKK family member SMG-1-8-9 in complex with a short peptide substrate (from UPF1) and a non-hydrolyzable analog of ATP, AMPPNP. This is an extension of the cryo-EM work that they did in 2019 on apo SMG-1-8-9 (Gat et al., 2019). However, this study is in fact more interesting than the previous work because it captures the SMG1 kinase in a state with its substrate peptide bound, frozen in time just before the actual act of phosphorylation. Importantly, the authors do an extensive characterization of the peptide recognition sequence, and reveal the chemical compatibility of the LSQ sequence and its derivatives for phosphorylation by SMG1. They compare the binding site of SMG1 to other members of the PIKK family and explain the observed differences between them in terms of substrate recognition. Overall, we believe that this work is novel and interesting. In general, the claims are supported by solid data, and the structure is validated with substantial biochemical work. Although we do not think additional experimental work is needed, below we make the following suggestions for revisions to strengthen the manuscript.Essential revisions:1) The structure and its interpretation seem plausible. However, we wonder if using an 11-amino acid peptide as a substrate captures all the contacts relevant for the kinase specificity that the authors are interested in. Is it possible that the full UPF1 molecule makes tertiary contacts with the kinase that are important for specificity? The authors should consider this possibility and explain why they can rule this out.

All the published in vivo and in vitro data at this point in time point to short phosphorylation motifs in the UPF1 unstructured regions as the determinants for recognition. Consistently, our biochemical assays (Figure 3B and Figure 1—figure supplement 1C) show that the peptide we used in the structural analysis recapitulates the specificity of SMG1 towards UPF1 SQ phosphorylation motifs. So far, we have not been able to detect the involvement of any other region of UPF1 in our biochemical studies or cryo-EM data. However, we do not rule out that there may be additional (and possibly regulated) interactions with the UPF1 structural region that may play a role in the crowded cellular environment and in the context of large NMD complexes, and indeed we are currently pursuing these studies. We have clarified this by adding a sentence at the end of the Conclusions, stating: “Whether phosphorylation of full-length UPF1 by SMG1 involves additional elements of recognition and/or additional levels of regulation will be a subject for future studies.”

2) Related to the above point, the authors state that "there are no extensive interactions between SMG1 and the residues preceding or following the LSQ motif in our structure." It is not clear how they can conclude this from their structure. If this is known from other work, they should cite it.

This statement was meant to describe the observations we make in our structure: The N- and C-terminal residues of the LSQ peptide do not show ordered density in our reconstruction, indicative of lack of extensive interactions with these residues (otherwise they would be ordered). This interpretation is supported by the assay in Figure 4—figure supplement 1B showing no significant effect of changing residues outside of the LSQ motif (in contrast to the prominent effect for changing residues within the motif itself). We have clarified that this is an observation by changing the text to: “we do not observe extensive interactions between SMG1 and the peptide residues preceding or following the LSQ motif in our structure. Consistently, we did not detect a marked effect on phosphorylation in a time course experiment where we changed the residue at position +2 of UPF1-LSQ (Figure 4—figure supplement 1B).”

3) The authors seem to have side-stepped the other 3D classes that were generated during data processing. There are at least two classes (center top and center bottom in Figure 1—figure supplement 3) that show extra density that is not present in the final reconstruction presented. That density appears to extend from SMG8's stalk and reach towards the FRB of SMG1. According to Li et al., 2019), this extra density is likely the kinase inhibitory domain (KID) of SMG8. This density also appeared in the authors' previous map of the apo SMG-1-8-9 complex (EMDB 10348), but they avoided discussing it in their previous manuscript (Gat et al., 2019). If the authors feel that delving into this would be too speculative, adding a label and a note to point out what they represent would be helpful.

The described extra density indeed likely corresponds to the less well resolved, unmodelled C-terminal half of SMG8 (compare Figure 1A). We have now highlighted these classes and indicated a potential regulatory function in the figure caption (Figure 1—figure supplement 3). We refrain from further speculation on the precise function of this part of SMG8 in the context of the data presented here.

4) A discussion regarding nucleotide binding to SMG9 and its effects on SMG8 binding and SMG1 catalytic activity is lacking. The authors in their previous work revealed a GTP/GDP nucleotide binding pocket in *C. elegans* SMG1 (Li et al., 2017), and then realized that human SMG9 co-purifies with bound ATP instead of GTP (Gat et al., 2019). Can human SMG1 also associate with GTP/GDP? In Figure 1—figure supplement 4 of this manuscript, the authors talk extensively about the adaptations for adenine vs guanine binding. This topic appears to be rather confusing in the field, so please include a short discussion that deals with the nucleotide-binding matter, including the mutagenesis study from another SMG-1-8-9 structure paper by Li et al., 2019.

The crystal structure of a *C. elegans* SMG8-SMG9 heterodimer purified from insect cells was initially solved as an apo-complex as well as bound to GDP following a crystal soaking experiment (Li et al., 2017). These soaking experiments had been guided by the overall G-domain fold of SMG9. More recently, mass spectrometry experiments using SMG1-8-9 as well as SMG8-9 complexes purified from HEK 293T cells revealed that SMG9 co-purifies bound to ATP, not with GTP (Gat et al. 2019). The higher resolution structural analysis carried out in the present manuscript confirms the presence of ATP in SMG8-SMG9 and explains the adaptions in the SMG9 G-fold making it indeed recognize an adenine base rather than a guanine base (Figure 1—figure supplement 4). Although uncommon, this is actually not the first instance where ATP binding to a G-fold domain has been observed. The speculation presented in the other SMG1-SMG8-SMG9 structure published last year (PMID: 31729466) was based on the (with hindsight wrong) assumption that SMG9 binds GTP from the earlier Li et al., paper (PMID: 28389433). While we would keep the majority of the details of ATP binding to the SMG9 G fold domain in the supplementary material (not to detract from the main take-home message of this paper), we have now added this paragraph: “The local resolution of around 3 Å allowed us to model SMG9-bound ATP in the reconstructed density, revealing the molecular basis for how the adenosine nucleotide is recognized by this unusual G-fold domain (Figure 1B and D, Figure 1—figure supplement 4). Briefly, the G4 and G5 motifs responsible for the recognition of the base have rearranged to preferentially bind an adenine base rather than a guanine (Figure 1—figure supplement 4).”

We have no evidence that human SMG1 kinase can associate with GTP/GDP, and believe this is very unlikely. We would defer from discussing and speculating on the impact of ATP-bound SMG9 on SMG1 activity in this paper, as this is not what the focus of this paper is about. In this paper, we focus on substrate specificity. Understanding the regulation of catalytic activity is beyond the scope of this manuscript.

5) The methods are not sufficiently detailed, and it would be challenging to reproduce the authors' work by simply following them.

We have expanded the respective Materials and methods section to improve clarity and included sufficient details to allow to reproduce this work.

6) Figure 3A and Figure 3 legend, particularly the statement that sequence logo letter size reflects the frequency of occurrence is not technically correct. True sequence logos use bit scores that typically do not reflect the frequency of occurrence but rather indicate the information content (in bits) that is being provided by each residue in that position in terms of the informational entropy content in each position of a motif, at least as originally described by Tom Schneider and Mike Stephens in 1990. A good review of the concept is found in Crooks et al., 2004. (http://www.genome.org/cgi/doi/10.1101/gr.849004). The authors should clarify how they are using the sequence logos here.

We have clarified this and modified the sentence to more precisely describe the meaning of the sequence logo: "Sequence logo derived from an alignment of all SQ motifs present in human UPF1 with the respective residue positions indicated. The heights of single letters correspond to the observed frequency at that position and the overall height of a stack of letters indicates the level of conservation (Crooks et al., 2004)."

7) Regarding the data in Figure 3B described in subsection “Crucial recognition of a glutamine residue at +1 position of the UPF1 consensus motif”, the claims of "peptide library" and "systematically" in the text go a bit beyond the data, since they are really looking at a small series of peptides, not a complete library of all residues in that position. The data is solid, and there is no need to do anything more, other than tone down the claims of this being a 'library'. We think the use of the term 'library' in Figure 4 and its description is fine, since in that case the authors really are looking at the set of all possible X-SQ sites that exist in UPF1.

We have changed this section to: "To validate the importance of this residue, we performed a mass spectrometry-based phosphorylation assay using a series of peptides based on UPF1-LSQ. We changed the residue at position +1 in the UPF1-LSQ peptide to test the effect of different side chain properties on phosphorylation."

8) The authors may want to expound a bit more on the similarity and differences in the hydrophobic cage between SMG1 and other PIKKs, particularly the Leu to Glu activation loop substitution seen in mTor (Figure 2—figure supplement 1B). This is important because the optimal kinase motif for mTor is NOT SQ as it is in ATM, ATR and DNA-PK, but rather more like SF or SP (see Figure 2 in Hsu et al., 2011).

We have included two sentences to stress this aspect and make it more easily understandable: "While the geometry of the hydrophobic cage is highly similar between SMG1 and CtTel1^ATM^, it deviates in mTOR due to the described Leu to Glu substitution. Indeed, mTOR has been found to prefer small or non-polar residues at position +1 of its phosphorylation consensus motif (Hsu et al., 2011)."

9) Subsection “Preferred recognition of a leucine residue at -1 position of the UPF1 consensus motif” and Figure 4C- presumably the point the authors are making is that C-terminal SQ motifs are better phosphorylated than the N-terminal ones because they are better matches to the hydrophocis-Ser-Gln motif. Perhaps they can state this conclusion a bit more clearly.

We have addressed this in the respective sentence: "Interestingly, further analysis of the final time points in the time course phosphorylation experiment showed that the SQ motifs that carry rather hydrophobic residues at the -1 position (and are therefore more efficiently phosphorylated) reside exclusively in the UPF1 C-terminus (Figure 4C, Figure 4—figure supplement 2)."

10) Similarly, it appears from Figure 2C that the Gln side chain is actually helping stabilize the activation segment conformation. Is this correct, and if so, shouldn't this be alluded to somewhere in the Results section?

We cannot conclude that there is a change in activation segment conformation upon substrate binding when comparing our reconstructions of apo- and substrate-bound SMG1 (see Figure 1—figure supplement 5). While the density of the substrate-bound active site appears more ordered compared to the apo reconstruction (Figure 1—figure supplement 5A vs. B), this is likely due to the overall increased quality of the map and we cannot objectively judge whether binding of Gln1079 stabilizes the conformation of the SMG1 hydrophobic cage.

11) Regarding the catalytic mechanism, as the authors have a (not so common) ternary complex protein kinase crystal structure containing both a peptide substrate and a nucleotide substrate, it would be nice to say a bit more about the relevance to potential phosphoryl transfer transition states. There has been a classical discussion in the field among physical organic chemists and enzymologists about whether phosphoryl transfers involving monoesters like the γ phosphate of ATP are more associative or dissociative. High resolution structures have been used to inform these discussions. For examples, please see: PMID:9408938, PMID:21513457, PMID:25399640

In the manuscript, we focus on phosphorylation motif recognition and tried avoiding making statements on detailed phosphoryl transfer mechanisms on purpose. The reason is that our cryo-EM structure was solved using a phosphate-based buffer. This may lead to depletion of a good portion of free magnesium ions from the buffer solution and potentially alter the precise conformation of the nucleotide in the active site. With the experimental caveat in mind, we do not feel comfortable in making strong in-depth statements on the chemistry of phosphoryl transfer transition states. However, we have taken the comment on board, and now added a sentence to Figure 2—figure supplement 1 pointing out that: “The SMG1 kinase contains features that have previously been associated with a model preferring the presence of a "dissociative" transition state, such as a positive charge closely involved in β-phosphate coordination, namely K2155 (residue not shown) (Wang and Cole, 2014)."